# A Multifunctional Fluorescent Probe Based on 1,8-Naphthalimide for the Detection of Co$^{2+}$, F$^-$, and CN$^-$

Ping Li [†], Xian-Xian Ji [†], Ming-Yao Xu, Yu-Long Liu and Liu Yang *

Department of Chemistry, College of Arts and Sciences, Northeast Agricultural University, Harbin 150030, China; liping@neau.edu.cn (P.L.); jixianxian@neau.edu.cn (X.-X.J.); m15145558510@163.com (M.-Y.X.); liuyulong@neau.edu.cn (Y.-L.L.)
* Correspondence: yangliu@neau.edu.cn; Tel.: +86-451-55191442
† These authors contributed equally to this work.

**Abstract:** Cations and anions are indispensable resources for the development of nature and modern industry and agriculture, and exploring more efficient technology to monitor them is urgently needed. A multifunctional fluorescent probe based on 1,8-naphthalimide, N-(2-thiophenhydrazide)acetyl-4-morpholine-1,8-naphthalimide (TMN), was successfully designed and synthesized for the detection of Co$^{2+}$, F$^-$, and CN$^-$, with N-carboxymethyl-4-morpholine-1,8-naphthalimide and thiophene-2-carbohydrazide as starting materials. TMN displayed superior stability in MeCN with an "on–off" mode towards Co$^{2+}$, F$^-$, and CN$^-$ by the naked eye. The linear response ranges of TMN were 0–3 and 4–19 μM with a detection limit of 0.21 μM for detecting Co$^{2+}$, 0–5 and 5–22 μM with a detection limit of 0.36 μM for F$^-$, and 0–10 and 10–25 μM with a detection limit of 0.49 μM for CN$^-$. TMN could also recognize Co$^{2+}$, F$^-$, and CN$^-$ in real samples. Finally, the possible sensing mechanisms of TMN for detecting Co$^{2+}$, F$^-$, and CN$^-$ were deeply investigated. These results implied that TMN could be a potential chemosensor for monitoring metal cations and anions sensitively and selectively and could be used in real sample detection.

**Keywords:** 1,8-naphthalimide derivatives; fluorescence; on–off; Co$^{2+}$, F$^-$, and CN$^-$ detection; naked-eye visible

## 1. Introduction

The trace element cobalt has a vital relationship with vitamin B12, which is also called cobalamin, the only vitamin containing a metal element [1–4]. Cobalt is a component of vitamin B12, and its physiological function is also displayed by the action of vitamin B12 [5]. Vitamin B12 enters the stomach through the intestinal tract, while cobalt can prevent vitamin B12 from being destroyed by microorganisms in the intestinal tract. The efficacy of vitamin B12 will be reduced or even disappear without the participation of cobalt. The lack of cobalt will lead to the formation of diseases, such as anemia, Alzheimer's disease, and sexual dysfunction, accompanied by asthma, abnormal intraocular pressure, body weight loss, glaucoma, and cardiovascular disease [6–8]. Fluoride anion is a micronutrient for human growth and plays a vital role in the treatment of osteoporosis and tooth damage [9,10]. In addition, its compounds are widely used in fluorine-containing pesticides and rubber. The excessive intake of fluoride anion can cause adverse physiological effects and may lead to urolithiasis, fluorosis, acute gastritis, and even cancer, etc. [11,12]. The Environmental Protection Agency (EPA, Washington, DC, USA) proposed that the standards for the concentration of fluoride anion in drinking water be 2 mg L$^{-1}$ (non-enforceable) and 4 mg L$^{-1}$ (enforceable) [13]. Cyanide is one of the most useful anions and has been widely used in many fields, such as extraction, electroplating, synthetic fibers, metallurgy, the resin industry, and herbicides [14,15]. Cyanide is also used as a chemical warfare agent and even as material for terrorism [16]. In addition, cyanide is highly toxic and harmful to humans

and the environment, especially being lethal to humans at concentrations of 0.5–3.5 mg/kg (body weight) [17,18].

At present, many materials have been utilized for monitoring cations and anions, such as fluorescent chemosensors [19,20], electrochemical sensors [21], carbon quantum dots [22,23], biosensors [24,25], and so on. The methods are mainly divided into three categories, according to their principles: chromatography, biochemical detection methods, and fluorescence detection methods. Among them, fluorescence detection methods have the characteristics of low cost, easy operation, a fast signal response, and signal visualization compared with the other two methods [26]. Fluorescence detection methods rely on the specific interaction of fluorescent chemosensors with analysts to analyze anions and cations based on the enhancement or quenching of fluorescent signals, or the changes in fluorescence spectra [27].

1,8-naphthalimide is one of the traditional environment-sensitive dyes, which exists in a large donor–acceptor electron-conjugated system and is susceptible to light transitions, resulting in strong fluorescence [28]. 1,8-naphthalimide-based fluorescent probes have the characteristics of high fluorescence quantum yield, excellent light resistance, strong chemical and thermal stability, easy structural modification, a large Stokes shift, and moderate emission wavelength, making naphthalimide one of the most valuable fluorophores [29]. In recent years, 1,8-naphthalimide-based fluorescent probes have become a research hotspot around the world. They have been applied to ion recognition, cell imaging, the detection of small molecular substances in cells and the detection of cell cancer, clinical medicine, etc. [30].

In this work, a multifunctional fluorescent probe, N-(2-thiophenhydrazide)acetyl-4-morpholine-1,8-naphthalimide (TMN), was successfully designed and synthesized for the detection of $Co^{2+}$, $F^-$, and $CN^-$ in a MeCN solution, with N-carboxymethyl-4-morpholine-1,8-naphthalimide and thiophene-2-carbohydrazide as starting materials. The fluorescence intensities of TMN-quenching were observed under 365 nm of UV light in the presence of $Co^{2+}$, $F^-$, and $CN^-$. TMN displayed superior sensitivity and selectivity to cations and anions. The limits of detection (LOD) of TMN for detecting $Co^{2+}$, $F^-$, and $CN^-$ were extremely low. TMN could also recognize $Co^{2+}$, $F^-$, and $CN^-$ in real samples. Finally, the mechanisms of TMN for detecting $Co^{2+}$, $F^-$, and $CN^-$ were investigated by $^1H$ NMR titration.

## 2. Results and Discussion

### 2.1. Fluorescence Spectral Characteristics Studies

In order to investigate the solvation effect of TMN, eight solvents, including methanol, tetrahydrofuran (THF), acetone, dichloromethane, acetonitrile, dimethyl sulfoxide (DMSO), N,N-dimethylformamide (DMF), and ethanol were selected for fluorescence spectral experiments (Figure S1a). As the polarity of the solvents increased, the main emission of the fluorescence spectra of TMN ($10^{-5}$ M) was red-shifted, and the fluorescence intensity decreased. This phenomenon may have contributed to the ICT process occurring in the system. Furthermore, the fluorescence spectra of TMN in different ratios of acetonitrile and water were also studied. As shown in Figure S1b, the fluorescence intensity of TMN was quenched in the presence of water. Finally, pure acetonitrile was used as the solvent for the UV-Vis and fluorescence spectral characteristics studies.

### 2.2. The Spectral Properties of TMN for the Detection of $Co^{2+}$, $F^-$, and $CN^-$

In order to explore the sensing performance of TMN, five eq. of different metal ions were added to the MeCN solutions of TMN ($10^{-5}$ M). In the UV-Vis spectra, TMN had a strong absorption peak at 396 nm, while the absorbance was enhanced by 2.1 and 3.1 times, with $Fe^{2+}$ and $Fe^{3+}$ added, respectively. The addition of $Cu^{2+}$ also caused changes in the UV-Vis spectra, while the change caused by $Co^{2+}$ was weak (Figure 1a). In fluorescence spectra, the intensity of the added $Co^{2+}$ in TMN was visible to the naked eye and was quenched from 2326 to 1174 at 530 nm ($\lambda_{ex}$ = 396 nm), while the changes caused by $Fe^{2+}$,

$Fe^{3+}$, and $Cu^{2+}$ were weak (Figure 1b), combined with the results of the UV-Vis spectra. Therefore, $Co^{2+}$ was selected for subsequent detection. As shown in Figure 2a, only when $F^-$ or $CN^-$ were added to the solution did the absorbance of the TMN at 300 to 350 nm of the UV-Vis spectra significantly enhance. In the fluorescence spectra, the intensities of TMN added with $F^-$ or $CN^-$ decreased to varying degrees, which was consistent with the UV-Vis spectra of the TMN (Figure 2b). This phenomenon implied that TMN could serve as a fluorescent probe for the detection of $Co^{2+}$, $F^-$, and $CN^-$.

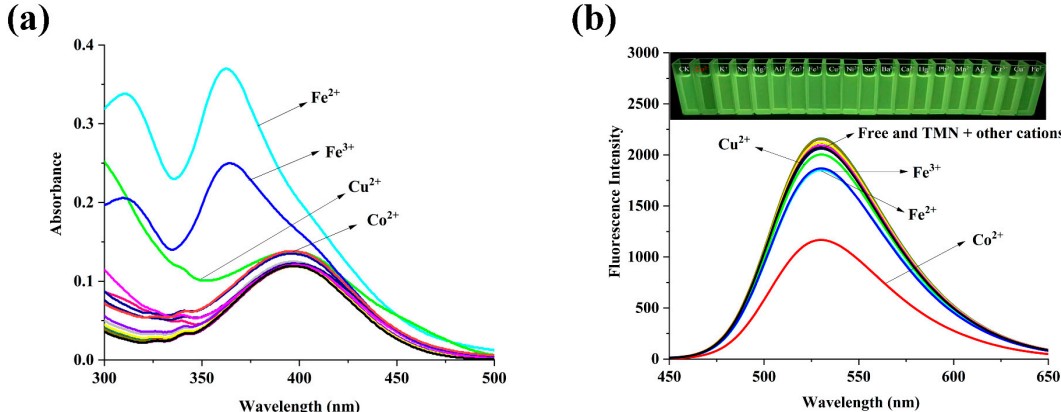

**Figure 1.** Effect of metal ions on UV-Vis (**a**) and fluorescence spectra (**b**) of TMN.

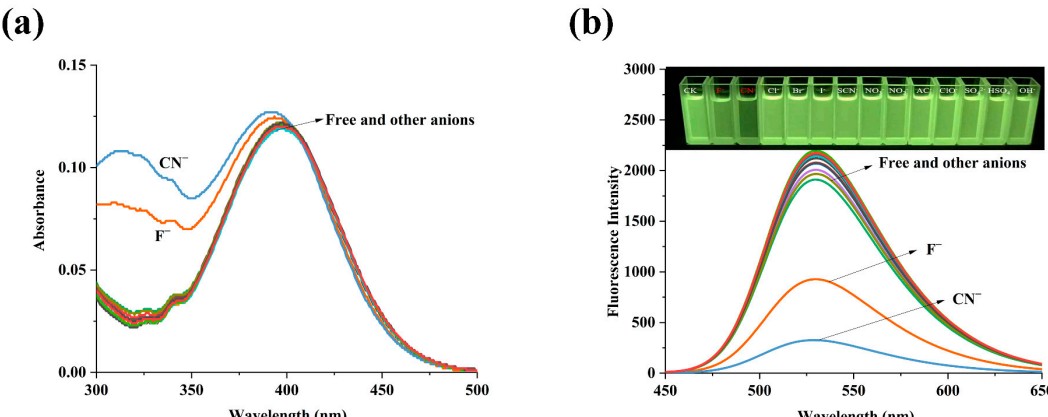

**Figure 2.** Effect of anions on UV-Vis (**a**) and fluorescence spectra (**b**) of TMN.

The interferences of other cations and anions on the recognition of $Co^{2+}$, $F^-$, and $CN^-$ by TMN ($10^{-5}$ M) in MeCN solutions were also studied. Five eq. of different metal ions or anions were first added to the solutions of TMN, and then five eq. of $Co^{2+}$, $F^-$, and $CN^-$ were added after the solutions stood for a period of time. The fluorescence intensities of TMN with other metal ions were nearly unchanged while they were quenched in the presence of $Co^{2+}$ (Figure 3a). $F^-$ was the same as $Co^{2+}$ (Figure 3b). For $CN^-$, the fluorescence intensity of TMN was almost unaffected by the addition of other anions, except for $SO_4^{2-}$, $HSO_4^-$, and $Ac^-$. Although these three ions interfered with it, their impact was relatively small and could be ignored (Figure 3c). Therefore, in the case of $F^-$ and $CN^-$ interfering with each other, there was almost no interference from other ions.

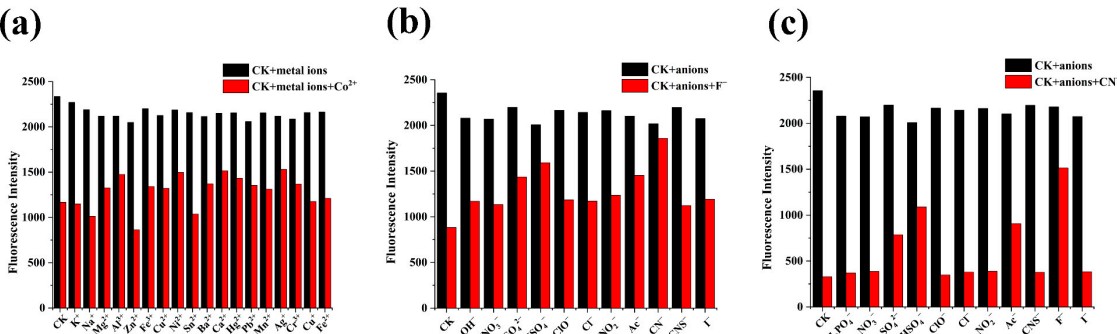

**Figure 3.** Interference selectivity of TMN towards $Co^{2+}$ (**a**), $F^-$ (**b**), and $CN^-$ (**c**) in the presence of other metal ions and anions.

In order to explore the fluorescence titration experiments of TMN, different amounts of $Co^{2+}$, $F^-$, and $CN^-$ were added to TMN to record the change in the fluorescence spectra. As shown in Figure 4a, the fluorescence intensities of TMN decreased with the increase in the concentration of $Co^{2+}$, which was from zero to five eq. (0 to 50 μM). When the concentration of $Co^{2+}$ reached 19 μM, the fluorescence intensity of TMN reached the minimum value and remained unchanged. The linear fitting equations of TMN for detecting $Co^{2+}$ were $y = 2147.06 - 65.20x$ ($R^2 = 0.99$) for 0 to 19 μM (Figure 4b). The LOD (limit of detection) of TMN for detecting $Co^{2+}$ was calculated to be 0.15 μM on the basis of the equation of LOD = $3\sigma/S$, where σ means the response standard deviation at the lowest concentration, and S is the slope of the calibration [31]. In Figure 4d, the fluorescence intensities of TMN decreased with the increasing concentrations of $F^-$, which were from zero to five eq. (0 to 50 μM). When the concentration of $CN^-$ reached 22 μM, the fluorescence intensity of TMN remained stable. The linear fitting equations of TMN for detecting $F^-$ were $y = 2107.47 - 56.96x$ ($R^2 = 0.99$) for 0 to 22 μM, and the LOD of TMN for detecting $F^-$ was 0.18 μM (Figure 4e). In Figure 4g, the fluorescence intensities of TMN also decreased with the increasing concentrations of $CN^-$, ranging from zero to seven eq. (0 to 70 μM). The linear fitting equations of TMN for detecting $CN^-$ were $y = 2282.81 - 81.90x$ ($R^2 = 0.99$) for 0 to 25 μM, with a LOD of 0.12 μM for detecting $CN^-$ (Figure 4h). Finally, we made a Stern–Volmer (S-V) plot to explore its quenching constants for $Co^{2+}$, $F^-$, and $CN^-$. As we all know, fluorescence quenching is usually divided into static quenching (SQ) and dynamic quenching (DQ), and both can be expressed by the S-V equation [32,33]:

$$F_0/F = 1 + K_{sv}[Q]$$

where $F_0$ and $F$ are the fluorescence intensities in the absence and presence of ions, respectively; [Q] is the ion concentration of the quenching agent; $K_{sv}$ is the quenching constant [34]. Since the fluorescence intensity ratio and ion concentration showed good linearity and conformed to the S-V equation ($R^2 = 0.99$), it was inferred that they were all static quenching [35,36], and the quenching constants of $Co^{2+}$, $F^-$, and $CN^-$ were 0.067, 0.060, and 0.189 $μM^{-1}$, respectively, which meant that $CN^-$ had the strongest quenching effect on TMN, followed by $Co^{2+}$ and $F^-$. Some reported probes for the detection of $Co^{2+}$, $F^-$, and $CN^-$ have been compared with TMN in Table 1 [37–42].

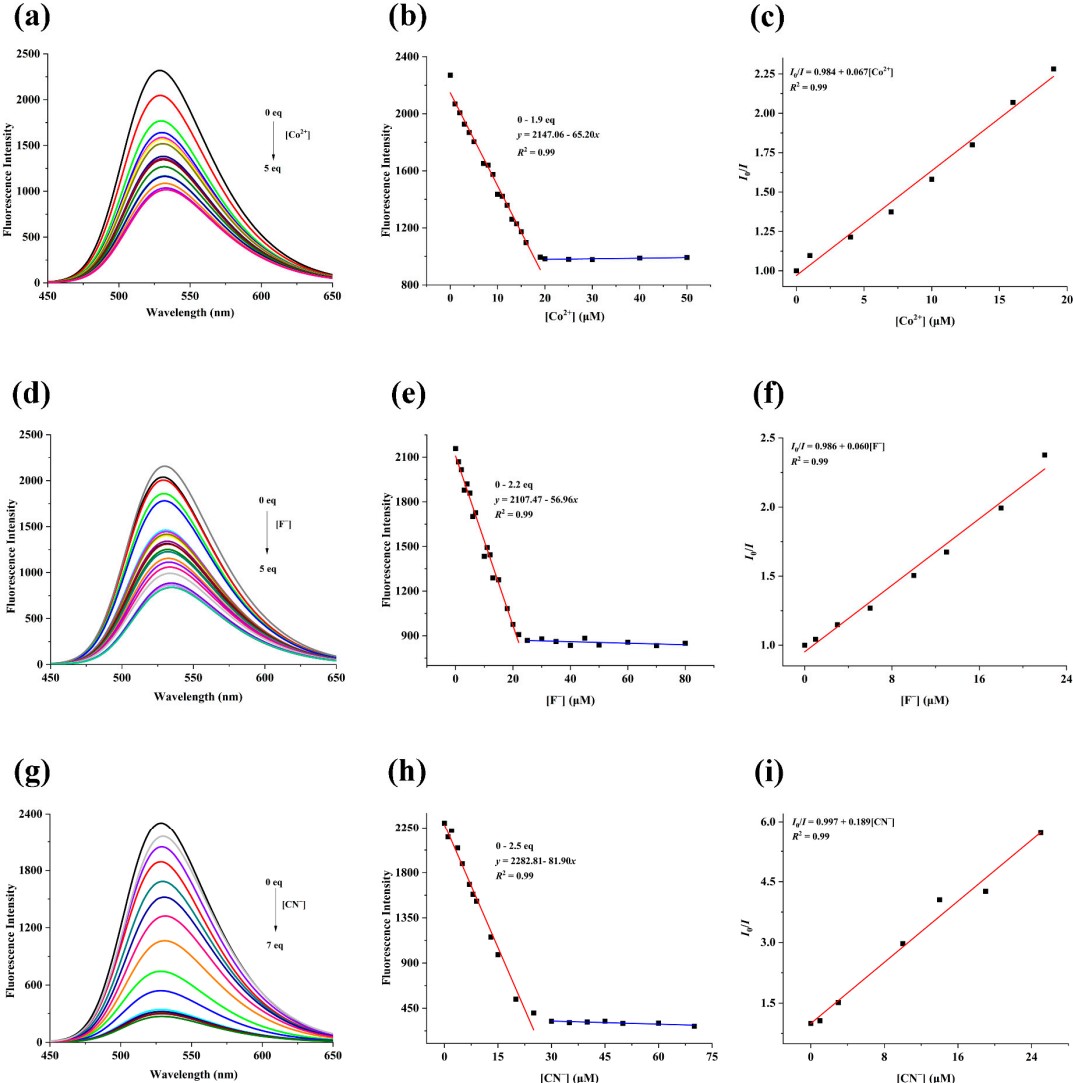

**Figure 4.** (**a**) Fluorescence spectra with different concentrations of ions, linear plots of fluorescence intensity, and ion concentrations and the Stern–Volmer plots. [Co$^{2+}$]: (**a–c**); [F$^-$]: (**d–f**); [CN$^-$]: (**g–i**).

Response time is an important factor for evaluating fluorescent probes in practical applications. The effect of the reaction time on the binding process of TMN to ions was also investigated. The results displayed that the fluorescence signal of TMN remained stable in the absence of any ions, showing good fluorescence stability. However, the fluorescence intensity of TMN decreased immediately in different degrees after adding Co$^{2+}$, F$^-$, and CN$^-$ to the solutions of TMN, respectively, reaching the minimum value within 10 *s*, and they remained constant for the following 60 min at 530 nm. This phenomenon implied that TMN had high reactivity with Co$^{2+}$, F$^-$, and CN$^-$ (Figure S2).

Finally, the binding ratios of TMN to Co$^{2+}$, F$^-$, and CN$^-$ were explored by Job's plot experiments. In Figure 5, the concentration ratios of the ions were from 0.1 to 0.9, where the total concentration of the TMN and ions was $1 \times 10^{-5}$ M. The maxima peaks were all at 0.5, which implied that the ratio of the interaction between the TMN and Co$^{2+}$, F$^-$, or CN$^-$ was 1:1.

**Table 1.** Comparison of TMN with other reported probes for the detection of $Co^{2+}$, $F^-$, and $CN^-$.

| Probes | Solution System | LOD | Liner Ranges | Applications | Ref. |
|---|---|---|---|---|---|
| 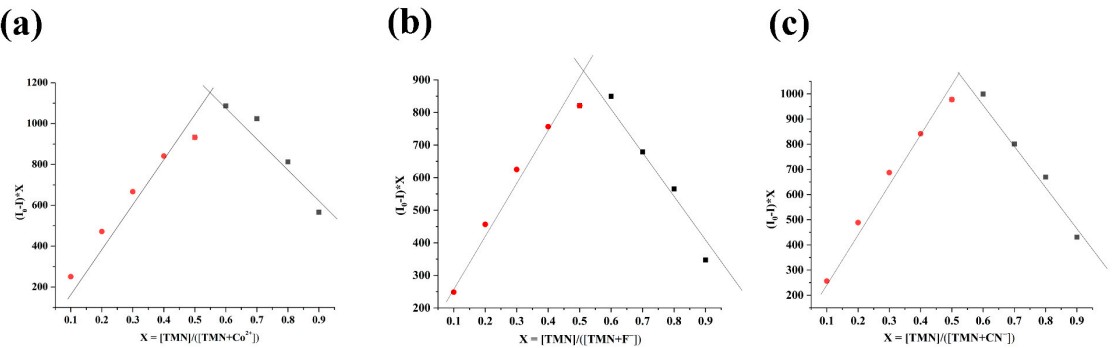 | EtOH/$H_2O$ ($Co^{2+}$) | $4.55 \times 10^{-3}$ μM | 0–0.5 μM | Cell imaging | [37] |
| | MeOH/$H_2O$ ($Co^{2+}$) | 0.21 μM | 0–12.5 μM | Cell imaging | [38] |
| | DMF/PBS ($F^-$) | 120 μM | 120 μM–1.5 mM | No statement | [39] |
| | DMSO ($F^-$) | 1.096 μM | 0–1 mM | No statement | [40] |
| | DMSO/$H_2O$ ($CN^-$) | 0.70 μM | 0–30 μM | Cell imaging | [41] |
| | DMSO/HEPES ($CN^-$) | 0.79 μM | 0–50 μM | Live animal imaging | [42] |
| | MeCN | 0.15 μM ($Co^{2+}$), 0.18 μM ($F^-$), 0.12 μM ($CN^-$) | 0–19 μM ($Co^{2+}$), 0–22 μM ($F^-$), 0–25 μM ($CN^-$) | Real water samples detection | TMN |

**(a)**     **(b)**     **(c)**

**Figure 5.** The Job's plot of **TMN** for $Co^{2+}$ (**a**), $F^-$ (**b**), and $CN^-$ (**c**).

*2.3. The Mechanism of TMN for the Detection of $Co^{2+}$, $F^-$, and $CN^-$*

To further explore the mechanism of TMN for detecting $Co^{2+}$, $^1H$ NMR titration analysis was carried out in a DMSO-$d_6$ solution. Due to the difference in the ligands and the electron arrangement of cobalt $3d_7$, Co(II) had a low-spin state, a high-spin state, and the coexistence of both, and low-spin Co(II) could be converted to low-spin $Co^{3+}$ [43]. The two could be distinguished by $^1H$ NMR spectra, with low-spin Co(II) complexes showing wide peaks and small paramagnetic shifts, high-spin Co(II) complexes showing narrow peaks and large paramagnetic shifts, and low-spin Co(III) complexes with diamagnetic and no paramagnetic shifts, so the Co(III) complexes in Figure 6 were in a low-spin state [44,45]. Since the chemical shift and number of protons of TMN were consistent with the $^1H$ NMR spectra of the TMN, indicating that the active H of TMN had no interaction with $Co^{2+}$, the mechanism of $Co^{2+}$ detection may be that the two acyl groups of TMN were coordinated with $Co^{2+}$, Co(II) was oxidized, and the reduced ligand changed its fluorescence spectrum [46–48] (Scheme 1).

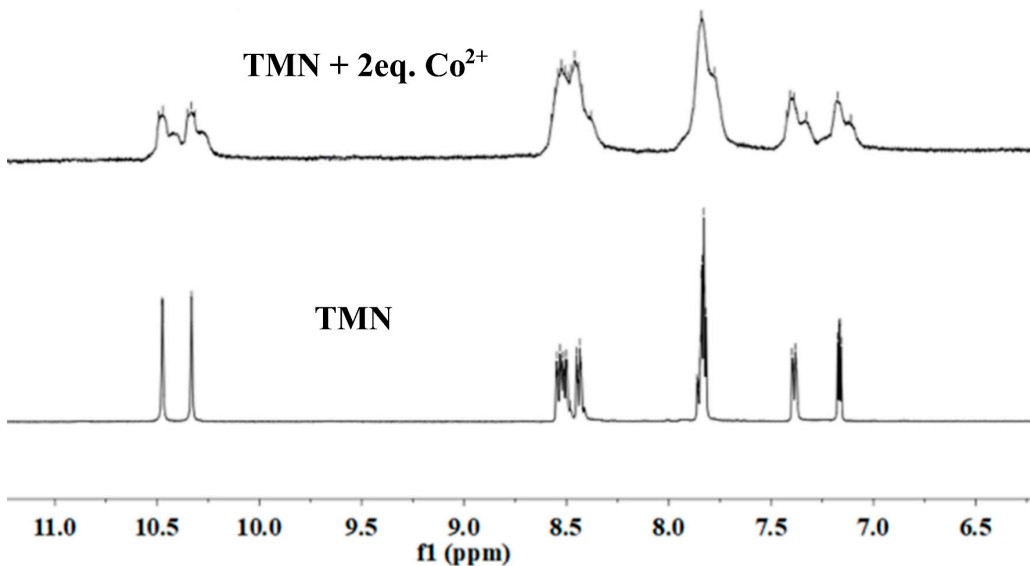

**Figure 6.** The $^1$H NMR titrations of TMN with Co$^{2+}$ in DMSO-$d_6$.

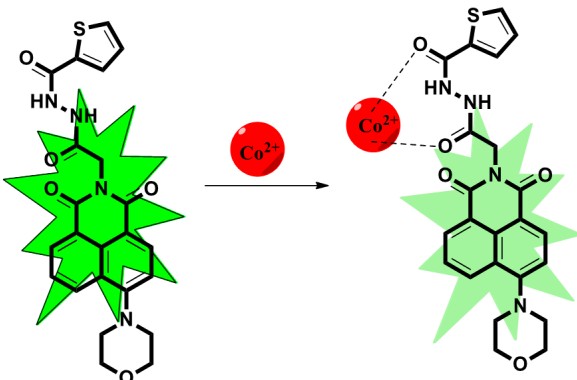

**Scheme 1.** The possible mechanism of TMN for the detection of Co$^{2+}$.

In Figure 7a, the two spikes at 10.2–10.5 ppm were two NH proton peaks, and with the addition of F$^-$, both signals shifted to the upper field, where one moved to 9.6 ppm and disappeared after becoming wider and smaller, and the other one moved to the 9.1 ppm spike, and the appearance of the next small peak may be due to intramolecular hydrogen bonding [49]. The new peaks that appeared at 6.7–8.0 ppm may be due to the upper field displacement of aromatic CH [50]. Meanwhile, a new weak proton signal was found at 15.56 ppm, which confirmed the presence of the dimer, [HF$_2$]$^-$ [51–53], implying deprotonation between the TMN and F$^-$. In addition, a similar interaction mechanism between TMN and CN$^-$ was also investigated by $^1$H NMR titration experiments (Figure 7b). Different amounts (one to five eq.) of TBACN were sequentially added to the DMSO-$d_6$ solution of TMN. When one eq. of CN$^-$ was added, the NH proton signal at 10.33 ppm immediately disappeared, and the other NH proton signal moved to the upper field site at 9.1 ppm, clearly indicating that deprotonation occurred [54,55]. Meanwhile, the addition of CN$^-$ destroyed the original molecular structure, and the proton peak that partially attached to the aryl group was displaced to the upward field, showing a new peak between 6.7 and 7.5 ppm [56]. The mechanism of TMN for F$^-$ or CN$^-$ is proposed in Scheme 2. Therefore, TMN could serve as a highly multifunctional fluorescent probe for the detection of Co$^{2+}$, F$^-$, and CN$^-$.

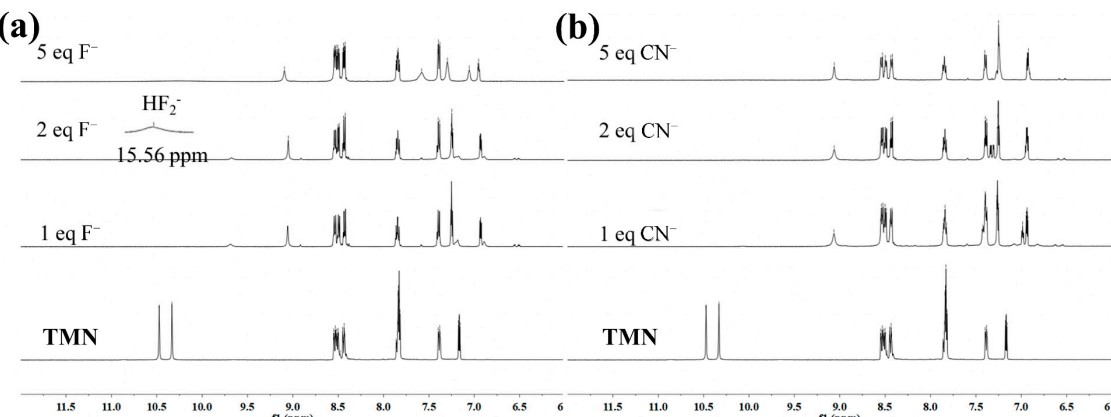

**Figure 7.** $^1$H NMR titration plot of TMN with F$^-$ (**a**) and CN$^-$ (**b**) in DMSO-$d_6$.

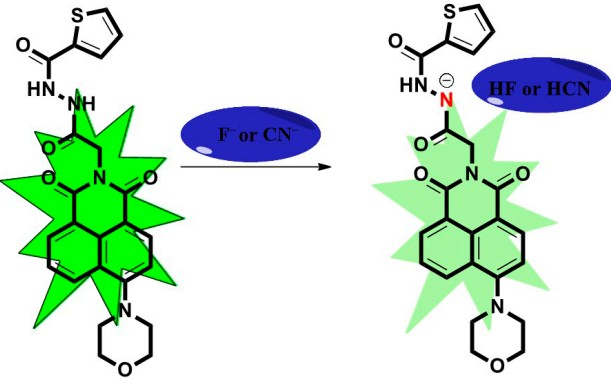

**Scheme 2.** The possible mechanism of TMN for the detection of F$^-$ or CN$^-$.

### 2.4. Detecting Co$^{2+}$, F$^-$, and CN$^-$ in Real Samples

In order to further explore the practical application of TMN, different water samples were collected to detect Co$^{2+}$, F$^-$, and CN$^-$ in actual water samples, and the standard addition method was introduced in this work. The final concentrations of Co$^{2+}$, F$^-$, and CN$^-$ in the water samples were 3.00 μM, 6.00 μM, and 9.00 μM, and the data are listed in Table 2. The recoveries of TMN for detecting Co$^{2+}$, F$^-$, and CN$^-$ all ranged from 98.56% to 110.67%, and all of the RSD values were within 3.67%. These data indicated that TMN could be a multifunctional fluorescent probe for monitoring Co$^{2+}$, F$^-$, and CN$^-$ in real samples.

**Table 2.** Results of detecting Co$^{2+}$, F$^-$, and CN$^-$ in real water samples.

| TMN | Add (μM) (Co$^{2+}$) | Found (μM) (Co$^{2+}$) | Recovery (%) (Co$^{2+}$) | RSD[b] (%) (Co$^{2+}$) |
|---|---|---|---|---|
| | 3.00 | 3.27 ± 0.04 | 109.00 ± 1.33 | 1.22 |
| Songhua River | 6.00 | 5.94 ± 0.02 | 99.00 ± 0.33 | 0.34 |
| | 9.00 | 9.19 ± 0.05 | 102.11 ± 0.56 | 0.54 |
| | 3.00 | 3.30 ± 0.06 | 110.00 ± 2.00 | 3.30 |
| Tap water | 6.00 | 6.31 ± 0.03 | 105.17 ± 0.50 | 0.48 |
| | 9.00 | 9.38 ± 0.06 | 104.22 ± 0.67 | 0.52 |

**Table 2.** *Cont.*

| TMN | Add (μM) (F$^-$) | Found (μM) (F$^-$) | Recovery (%) (F$^-$) | RSD[b] (%) (F$^-$) |
|---|---|---|---|---|
| Songhua River | 3.00 | 3.14 ± 0.04 | 104.67 ± 1.33 | 1.92 |
| | 6.00 | 6.28 ± 0.05 | 104.67 ± 0.83 | 3.67 |
| | 9.00 | 9.16 ± 0.03 | 101.78 ± 0.33 | 0.79 |
| Tap water | 3.00 | 3.32 ± 0.04 | 110.67 ± 1.33 | 3.48 |
| | 6.00 | 6.30 ± 0.03 | 105.17 ± 0.50 | 0.72 |
| | 9.00 | 9.29 ± 0.05 | 104.22 ± 0.56 | 1.08 |
| **TMN** | **Add (μM) (CN$^-$)** | **Found (μM) (CN$^-$)** | **Recovery (%) (CN$^-$)** | **RSD[b] (%) (CN$^-$)** |
| Songhua River | 3.00 | 3.18 ± 0.03 | 106.00 ± 1.00 | 0.63 |
| | 6.00 | 6.25 ± 0.03 | 104.17 ± 0.50 | 0.48 |
| | 9.00 | 8.87 ± 0.05 | 98.56 ± 0.56 | 0.56 |
| Tap water | 3.00 | 3.25 ± 0.03 | 108.33 ± 1.00 | 1.12 |
| | 6.00 | 6.33 ± 0.04 | 105.50 ± 0.67 | 0.47 |
| | 9.00 | 9.42 ± 0.06 | 104.67 ± 0.67 | 0.64 |

Average of three repeated measurements of $Co^{2+}$, F$^-$, and CN$^-$. b: RSD means relative standard deviation.

## 3. Experimental Section

### 3.1. Materials and Physical Measurements

All the reactants and solvents utilized in this work were commercially available with no further purification. KCl, NaCl, $MgCl_2$, $AlCl_3$, $ZnCl_2$, $FeCl_3$, $CaCl_2$, $NiCl_2$, $SnCl_2$, $BaCl_2$, $CuCl_2$, $HgCl_2$, $Pb(NO_3)_2$, $MnCl_2$, $AgNO_3$, $CrCl_3$, CuCl, $FeCl_2$, and $CoCl_2$ were purchased from Sigma Aldrich. Anions of $H_2PO_4^-$, OH$^-$, $NO_3^-$, $SO_4^{2-}$, $HSO_4^-$, ClO$^-$, Cl$^-$, $NO_2^-$, Ac$^-$, CNS$^-$, Br$^-$, I$^-$, F$^-$, and CN$^-$ were purchased from Sigma-Aldrich, which were all tetrabutylammonium (TBA) salts. Methanol (MeOH), ethanol (EtOH), acetonitrile (MeCN), dimethyl sulfoxide (DMSO), N,N′-dimethylformamide (DMF), dichloromethane, acetone, and tetrahydrofuran (THF) were also purchased from Sigma-Aldrich. High-resolution mass spectrometry (HRMS) was carried out on an Agilent 6224. The [1]H NMR and [13]C NMR spectra of samples were obtained through a Bruker AVANVE 400 MHz system (Bruker, Germany). The fusion points were recorded on a Shanghai Inesa melting point apparatus (WRS-3) without correction. UV-Vis spectra were measured on a Shimadzu UV-2700 spectrophotometer. IR spectra were recorded on an Alpha Centaurt FT/IR spectrophotometer over a wavenumber ranging from 4000 to 400 cm$^{-1}$ using KBr pellets. Fluorescence spectra were gathered on an LS-55 using a xenon lamp and quartz carrier at an ambient temperature.

### 3.2. Synthesis of N-(2-thiophenhydrazide)acetyl-4-morpholine-1,8-naphthalimide (TMN)

The syntheses of compounds **1** and **2** [57,58] are introduced in the supporting information, and the relative spectra are listed in Figures S3–S8. Compound **2** (5 mmol) was reacted with thiophene-2-carbohydrazide by the coupling reagents 1-ethyl-3-(3-dimethylaminopropyl) carbodiimide (EDC, 1.1 eq.), the base N,N-diisopropylethylamine (DIEA, 2 eq.), and hydroxybenzotrizole (HOBt, 1.2 eq.) in dry DMF at 150 °C for 12 h. The reaction was then quenched by adding water, and the desired product was precipitated from the reaction mixture, which was further filtered and dried. The mixture was purified by column chromatography on silica gel eluted with $CH_2Cl_2$/MeOH (V/V, 15/1) to obtain yellow TMN (Scheme 3); yield: 80%. m.p.: 220.6–221.3 °C. All spectra of the structural characterization of the compound TMN are listed in Figure S9–S12. FT-IR (KBr) cm$^{-1}$: 3273 (v N-H), 2995, 2872 (v C-H), 1693, 1660 (v C=O). [1]H NMR (DMSO-$d_6$, TMS, 400 MHz, ppm) δ 10.47 (d, J = 1.4 Hz, 1H), 10.33 (s, 1H), 8.57–8.40 (m, 3H), 7.83 (td, J = 5.9, 4.9, 3.4 Hz, 3H), 7.40–7.38 (m, 1H), 7.17 (dd, J = 4.9, 3.8 Hz, 1H), 4.79 (s, 2H), 3.92 (t, J = 4.5 Hz, 4H), 3.25 (t, J = 4.5 Hz, 4H). [13]C NMR (DMSO-$d_6$, TMS, 100 MHz, ppm): 167.02, 163.80, 163.23, 160.85, 156.12,

137.70, 132.78, 132.08, 131.25, 129.97, 129.67, 129.42, 128.60, 126.55, 125.69, 122.79, 115.97, 115.51, 66.65, 53.51, 41.42; HRMS (ESI) was calculated for $C_{23}H_{20}N_4O_5S$ ([M+H]$^+$) 465.1227, found 465.1217.

**Scheme 3.** Synthetic route of TMN.

### 3.3. General Preparation for the Spectral Experiments

TMN was prepared to the concentration of $10^{-4}$ M by MeCN as the stock solution. Different metal salts and TBA salts were prepared to the concentration of $10^{-2}$ M by deionized water. Next, 1 mL of the TMN stock solution was diluted to the concentration of $10^{-5}$ M with the MeCN solution in a 10 mL volumetric flask. Furthermore, 5 eq. (50 μL) of different cations and anions were added to $10^{-5}$ M of TMN solutions by ultrasonication to form homogeneous solutions. Finally, different concentrations of $Co^{2+}$, $F^-$, and $CN^-$ were titrated to $10^{-5}$ M of TMN to study the ability of TMN to detect cations and anions.

### 3.4. Detecting $Co^{2+}$, $F^-$, and $CN^-$ in Real Samples

Tap water was collected from the Northeast Agricultural University (Harbin, China). The Songhua River water and tap water were filtered with a 0.45 μm microporous membrane for use. The standard addition method was been chosen for sensing $Co^{2+}$, $F^-$, and $CN^-$ in real samples. The added concentrations of $Co^{2+}$, $F^-$, and $CN^-$ were 30, 60, and 90 μM, respectively, using a 9:1 ratio of TMN to water.

## 4. Conclusions

In summary, a multifunctional fluorescent probe, TMN, based on 1,8-naphthalimide, was designed and synthesized for the detection of $Co^{2+}$, $F^-$, and $CN^-$. TMN displayed superior stability in MeCN with an "on–off" mode towards $Co^{2+}$, $F^-$, and $CN^-$ by the naked eye. The detection limits of TMN for detecting $Co^{2+}$, $F^-$, and $CN^-$ were 0.21 μM, 0.36 μM, and 0.49 μM, respectively. Meanwhile, the possible sensing mechanisms of TMN for detecting $Co^{2+}$, $F^-$, and $CN^-$ were deeply investigated by $^1$H NMR titration analysis. TMN could also recognize $Co^{2+}$, $F^-$, and $CN^-$ in real water samples. These results indicated that TMN could be a potentially multifunctional chemosensor for the sensitive and selective detection of metal cations and anions.

**Supplementary Materials:** The following supporting information can be downloaded at: https://www.mdpi.com/article/10.3390/inorganics11070265/s1, Figure S1: (a) Fluorescence response of TMN in different solvents; (b) Fluorescence response of TMN in solutions with different MeCN to water ratios, Figure S2: Fluorescence intensity of TMN in the absence and presence of $Co^{2+}$, $F^-$, and $CN^-$, Figure S3: IR spectrum of compound 1, Figure S4: $^1H$ NMR spectrum of compound 1, Figure S5: $^{13}C$ NMR spectrum of compound 1, Figure S6: IR spectrum of compound 2, Figure S7: $^1H$ NMR spectrum of compound 2, Figure S8: $^{13}C$ NMR spectrum of compound 2, Figure S9: IR spectrum of TMN, Figure S10: $^1H$ NMR spectrum of TMN, Figure S11: $^{13}C$ NMR spectrum of TMN, Figure S12: Mass spectrum $(M+H^+)$ of TMN compound.

**Author Contributions:** P.L. and X.-X.J. repeated the experiment multiple times and then used it for in-depth research content. M.-Y.X. used Origin and Excel to process the data. Y.-L.L. consulted related literature and collected background knowledge and theoretical support. L.Y. conceived and designed the experiment. All authors have read and agreed to the published version of the manuscript.

**Funding:** Thank you to the University Nursing Program for Young Scholars with Creative Talents in Heilongjiang Province, China (No. UNPYSCT-2020111), for this work.

**Data Availability Statement:** Not applicable.

**Conflicts of Interest:** The authors declare no competing financial interest.

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
