# Peer review of "A Multifunctional Fluorescent Probe Based on 1,8-Naphthalimide for the Detection of Co2+, F, and CN"

_inorganics, doi:10.3390/inorganics11070265_

Round 1

Reviewer 1 Report (New Reviewer)

The article " A multifunctional fluorescent probe based on 1,8-naphthalimide for the detection of Co2+, F− and CN−” describes a fluorescent method for the determination of Co2+, F− and CN− using 1,8-naphthalimide derivative. The work was done well, but there are some areas that require improvement, as outlined below:

1. Figure 3b and 3c do not show CN- and F- ions, respectively. They need to be added. CN- and F- ions have the same detection mechanism and can interfere with each other.

2. Include the impact of different anions on the measurement of cobalt ions.

3. The recognition mechanism for anions is based on probe deprotonation (Scheme 2). A detailed explanation is required for why the probe does not respond to the hydroxide ion. In acetonitrile, the hydroxide ion is a stronger base than CN- and F-.

4. “the fluorescence intensities of TMN after adding other anions were not nearly affected except that the interferences caused by HSO4−and CNS− could be ignored” A more detailed explanation is needed for why these anions can be ignored. For instance, Figure 3c suggests that it is impossible to quantitatively determine cyanide ions when they are present in equal concentrations with sulfate, hydrosulfate, and acetate.

5. How does adding water to acetonitrile affect the measurement of Co2+, CN-, and F- ions in real samples?

6. Section 3.4 in the experimental part is incomplete. What was the water content of the measured samples in acetonitrile when determining the ions?

7. The caption for Scheme 2 is incorrect.

8. Provide the detection limit values with one unit of measurement in Table 1

Author Response

Answers for questions

For Referee 1

The article " A multifunctional fluorescent probe based on 1,8-naphthalimide for the detection of Co2+, F- and CN-” describes a fluorescent method for the determination of Co2+, F- and CN- using 1,8-naphthalimide derivative. The work was done well, but there are some areas that require improvement, as outlined below:

  1. Figure 3b and 3c do not show CN- and F- ions, respectively. They need to be added. CN- and F- ions have the same detection mechanism and can interfere with each other.

Author reply: Thank you for your helpful suggestions. We have added relevant content.

Line 116: Therefore, in the case of F and CN interfering with each other, there was almost no interference from other ions.

Figure 3. Interference selectivity of TMN towards Co2+ (a), F (b) and CN (c) in the presence of other metal ions and anions.

  1. Include the impact of different anions on the measurement of cobalt ions.

Author reply: Thank you for your helpful suggestions. Due to the fact that cobalt ions are cations, competitive experiments should focus on the fluorescence response relative to other cations, such as Ngamdee et al., Meng et al., Ng et al., Wang et al., Liu et al. and Sang et al.

References:

Ngamdee, K.; Tuntulani, T.; Ngeontae, W. L-Cysteine modified luminescence nanomaterials as fluorescence sensor for Co2+: Effects of core nanomaterials in detection selectivity, Sensor. Actuat. B-Chem. 2015, 216, 150-158.

Meng, L.; Zhu, Q.; Yin, J.H.; Xu, N. Polyethyleneimine protected silver nanoclusters luminescence probe for sensitive detection of cobalt (II) in living cells, J. Photoch. Photobio. B. 2017, 173, 508-513.

Ng, Y.H.; Chin, S.F.; Pang, S.C.; Ng, S.M. Utilising the interface interaction on tris(hydroxymethy)aminomethane-capped carbon dots to enhance the sensitivity and selectivity towards the detection of Co(II) ions, Sensor. Actuat. B-Chem. 2018, 273, 83-92.

Wang, S., Bao, X., Gao, B.; Li, M. A novel sulfur quantum dot for the detection of cobalt ions and norfloxacin as a fluorescent “switch”, Dalton Trans. 2019, 48, 8288-8296.

Liu, Y.L.; Yang, L.; Li, L.; Guo, Y.Q.; Pang, X.X.; Li, P.; Ye, F.; Fu, Y. A New Fluorescent Chemosensor for Cobalt(II) Ions in Living Cells Based on 1,8-Naphthalimide, Molecules 2019, 24, 3093.

Sang, F.; Zhang, X., Shen, F. Fluorescent methionine-capped gold nanoclusters for ultra-sensitive determination of copper(II) and cobalt(II), and their use in a test strip, Microchim. Acta 2019, 186, 373.

  1. The recognition mechanism for anions is based on probe deprotonation (Scheme 2). A detailed explanation is required for why the probe does not respond to the hydroxide ion. In acetonitrile, the hydroxide ion is a stronger base than CN- and F-.

Author reply: Thank you for your helpful suggestions. OH- was included in the other anions mentioned in Figure 2, but according to the results, OH- did not have the same effect as F- and CN-. In my opinion, the reason for the color change caused by detection for TMN of F- and CN- is that the interaction of hydrogen bonds is greater than deprotonation. First, with the addition of ions TMN and hydrogen bonds between F- and CN- interact, deprotonation occurs as the ion concentration increases.

  1. “the fluorescence intensities of TMN after adding other anions were not nearly affected except that the interferences caused by HSO4- and CNS- could be ignored” A more detailed explanation is needed for why these anions can be ignored. For instance, Figure 3c suggests that it is impossible to quantitatively determine cyanide ions when they are present in equal concentrations with sulfate, hydrosulfate, and acetate.

Author reply: Thank you for your helpful suggestions. We have redescribed this section.

Line 113: For CN, the fluorescence intensity of TMN was almost unaffected by the addition of other anions, except for SO42, HSO4, and Ac. Although these three ions interfere with it, their impact was relatively small and could be ignored (Figure 3c).

  1. How does adding water to acetonitrile affect the measurement of Co2+, CN-, and F- ions in real samples?

Author reply: Thank you for your helpful suggestions. According to Figure S1, TMN has the highest fluorescence intensity in pure acetonitrile, but due to the toxicity of acetonitrile and the consideration that the real sample is a water sample, we use acetonitrile to water ratio of 9:1, in this case, although the fluorescence effect is not the best, but it is more suitable for testing in real samples.

  1. Section 3.4 in the experimental part is incomplete. What was the water content of the measured samples in acetonitrile when determining the ions?

Author reply: Thank you for your helpful suggestions. We have added relative contents in the manuscript.

Line 270: Tap water were collected from the Northeast Agricultural University (Harbin, China). The Songhua River water and tap water were filtered with a 0.45 μm microporous membrane for use. The standard addition method has been chosen for sensing Co2+, F and CN in real samples. The added concentrations of Co2+, F and CN were 30, 60, and 90 μM, respectively, using a 9:1 ratio of TMN to water.

  1. The caption for Scheme 2 is incorrect.

Author reply: Thank you for your helpful suggestions. It has been corrected.

Line 212: Scheme 2. The possible mechanism of TMN for the detection of F- (a) and CN- (b).

  1. Provide the detection limit values with one unit of measurement in Table 1.

Author reply: Thank you for your helpful suggestions. It has been corrected.

Table 1. Comparison of TMN with other reported probes for the detection of Co2+, F and CN.

Probes

Solution

system

LOD

Liner

ranges

Applications

Ref.

EtOH/H2O

(Co2+)

4.55 × 10−3 μM

0-0.5 μM

Cell imaging

[44]

MeOH/H2O

(Co2+)

0.21 μM

0-12.5 μM

Cell imaging

[45]

DMF/PBS

(F)

120 μM

120 μM-1.5 mM

No statement

[46]

DMSO

(F)

1.096 μM

0-1 mM

No statement

[47]

DMSO/H2O

(CN)

0.70 μM

0-30 μM

Cell imaging

[48]

DMSO/HEPES

(CN)

0.79 μM

0-50 μM

Live animal imaging

[49]

MeCN

0.15 μM (Co2+),

0.18 μM (F),

0.12 μM (CN)

0-19 μM (Co2+),

0-22 μM (F),

0-25 μM (CN)

Real water samples detection

TMN

Reviewer 2 Report (New Reviewer)

The text describes the development of a multifunctional fluorescent probe called TMN, based on 1,8-naphthalimide, for the detection of Co2+, F−, and CN− ions. TMN was also tested for its ability to detect Co2+, F−, and CN− ions in real samples, suggesting its potential application as a chemosensor for monitoring metal cations and anions in practical settings.

At lines 71-73: Authors write: “The fluorescence intensities of TMN quenched observed under 365 nm UV light in the presence of  Co2+, F− and CN−. “Could they explain why they mention here under 365 nm UV light, since this wavelength is not found used anywhere else in the text, more details.

In the Caption of Figure 7. “Effect of metal ions” should be “Effect of anions”.

Line 128: Please check the expression ”of the calibration and [36,37].”

Table 1, line 1 applicatio=applications;

The solvent acetonitrile appears abbreviated in the text in two (different) ways, if namely in lines 15,91,109,229 appears MeCN and in lines 70,260,263 appears CH3CN, please correct. It should be expressed the same in the whole manuscript.

Line 267: the title 3.4 of this line is extra.

Minor points: Line 72: quenched observed= quenched was observed; line 84: process occurred = process occurring; line 94: was week = was weak; line 150: concentrations of ion= concentrations of ions; line 196: investigeted = investigated.

The text describes the development of a multifunctional fluorescent probe called TMN, based on 1,8-naphthalimide, for the detection of Co2+, F−, and CN− ions. TMN was also tested for its ability to detect Co2+, F−, and CN− ions in real samples, suggesting its potential application as a chemosensor for monitoring metal cations and anions in practical settings.

At lines 71-73: Authors write: “The fluorescence intensities of TMN quenched observed under 365 nm UV light in the presence of  Co2+, F− and CN−. “Could they explain why they mention here under 365 nm UV light, since this wavelength is not found used anywhere else in the text, more details.

In the Caption of Figure 7. “Effect of metal ions” should be “Effect of anions”.

Line 128: Please check the expression ”of the calibration and [36,37].”

Table 1, line 1 applicatio=applications;

The solvent acetonitrile appears abbreviated in the text in two (different) ways, if namely in lines 15,91,109,229 appears MeCN and in lines 70,260,263 appears CH3CN, please correct. It should be expressed the same in the whole manuscript.

Line 267: the title 3.4 of this line is extra.

Minor points: Line 72: quenched observed= quenched was observed; line 84: process occurred = process occurring; line 94: was week = was weak; line 150: concentrations of ion= concentrations of ions; line 196: investigeted = investigated.

Author Response

Answers for questions

For Referee 2

The text describes the development of a multifunctional fluorescent probe called TMN, based on 1,8-naphthalimide, for the detection of Co2+, F-, and CN- ions. TMN was also tested for its ability to detect Co2+, F-, and CN- ions in real samples, suggesting its potential application as a chemosensor for monitoring metal cations and anions in practical settings.

  1. At lines 71-73: Authors write: “The fluorescence intensities of TMN quenched observed under 365 nm UV light in the presence of Co2+, F- and CN-. “Could they explain why they mention here under 365 nm UV light, since this wavelength is not found used anywhere else in the text, more details.

Author reply: To make it more intuitive and visible to the naked eye, we photographed it at 365 nm using a portable UV analyzer and showed it in Figures 1 and 2.

  1. In the Caption of Figure 7. “Effect of metal ions” should be “Effect of anions”.

Author reply: Thank you for your helpful suggestions. We have corrected “Effect of metal ions” to “Effect of anions”.

Line 107: Figure 2. Effect of anions on UV-Vis (a) and fluorescence spectra (b) of TMN.

  1. Line 128: Please check the expression ”of the calibration and [36,37].”

Author reply: Thank you for your helpful suggestions. We have rechecked and revised.

Line 128: The LOD (limit of detection) of TMN for detecting Co2+ was calculated to be 0.15 μM on the basis of the equation of LOD = 3σ/S, where σ means the response standard deviation at the lowest concentration and S is the slope of the calibration [36,37].

  1. Table 1, line 1 applicatio=applications;

Author reply: Thank you for your helpful suggestions. We have corrected “applicatio” to “applications”.

Table 1. Comparison of TMN with other reported probes for the detection of Co2+, F and CN.

Probes

Solution

system

LOD

Liner

ranges

Applications

Ref.

EtOH/H2O

(Co2+)

4.55 × 10−3 μM

0-0.5 μM

Cell imaging

[44]

MeOH/H2O

(Co2+)

0.21 μM

0-12.5 μM

Cell imaging

[45]

DMF/PBS

(F)

120 μM

120 μM-1.5 mM

No statement

[46]

DMSO

(F)

1.096 μM

0-1 mM

No statement

[47]

DMSO/H2O

(CN)

0.70 μM

0-30 μM

Cell imaging

[48]

DMSO/HEPES

(CN)

0.79 μM

0-50 μM

Live animal imaging

[49]

MeCN

0.15 μM (Co2+),

0.18 μM (F),

0.12 μM (CN)

0-19 μM (Co2+),

0-22 μM (F),

0-25 μM (CN)

Real water samples detection

TMN

  1. The solvent acetonitrile appears abbreviated in the text in two (different) ways, if namely in lines 15,91,109,229 appears MeCN and in lines 70,260,263 appears CH3CN, please correct. It should be expressed the same in the whole manuscript.

Author reply: Thank you for your helpful suggestions. We have checked the full text and kept it consistent.

Line 68: In this work, a multifunctional fluorescent probe, N-(2-thiophenhydrazide)acetyl-4-morpholine-1,8-naphthalimide (TMN) was successfully designed and synthesized for the detection of Co2+, F- and CN- in MeCN solution with N-carboxymethyl-4-morpholine-1,8-naphthalimide and thio-phene-2-carbohydrazide as starting materials.

Line 263: TMN was prepared to the concentration of 10-4 M by MeCN as the stock solution.

Line 265: Next, 1 mL of TMN stock solution was diluted to the concentration of 10-5 M with MeCN solution in 10 mL volumetric flask.

  1. Line 267: the title 3.4 of this line is extra.

Author reply: Thank you for your helpful suggestions.

Line 270: Tap water were collected from the Northeast Agricultural University (Harbin, China). The Songhua River water and tap water were filtered with a 0.45 μm microporous membrane for use. The standard addition method has been chosen for sensing Co2+, F and CN in real samples. The added concentrations of Co2+, F and CN were 30, 60, and 90 μM, respectively, using a 9:1 ratio of TMN to water.

  1. Minor points: Line 72: quenched observed= quenched was observed; line 84: process occurred = process occurring; line 94: was week = was weak; line 150: concentrations of ion= concentrations of ions; line 196: investigeted = investigated.

Author reply: Thank you for your helpful suggestions. We have made corresponding corrections.

Line 71: The fluorescence intensities of TMN quenching was observed under 365 nm UV light in the presence of Co2+, F- and CN-.

Line 84: This phenomenon may contribute to the ICT process occurring in the system.

Line 93: The addition of Cu2+ also caused the changes of UV-Vis spectra, while the change caused by Co2+ was weak (Figure 1a).

Line 153: Figure 4. (a) Fluorescence spectra with different concentrations of ions, linear plots of fluorescence intensity and ion concentrations and the Stern-Volmer plots. [Co2+]: a-c; [F-]: d-f; [CN-]: g-i.

Line 198: In addition, a similar interaction mechanism between TMN and CN- was also investigated by 1H NMR titration experiments (Figure 7b).

Round 2

Reviewer 1 Report (New Reviewer)

Dear Authors,

I am satisfied with your responses to my comments 

This manuscript is a resubmission of an earlier submission. The following is a list of the peer review reports and author responses from that submission.

Round 1

Reviewer 1 Report

Please, see attached document.

Extensive editting of English language along all the Manuscript is required. 

The construction of sentences can be improved in many aspects to clarify their meaning and facilitate their reading. Typographical errors have also been detected (see Table 1 for example).

Reviewer 2 Report

The manuscript "A multifunctional fluorescent probe based on 1,8-naphthalimide for the detection of Co2+, F− and CN−" describes the synthesis and study of properties of new fluorescent probe, a representative of 1,8-naphthalimide calss, which is sensitive towards cobalt(II) and fluoride and cyanide anions in acetonitrile. The paper is interesting, however, there are some point that need additional attention from my point of view:

1. It is quite unusual that Co2+ added to the ligand alters its fluorescence intensity, but do not change its UV-Vis spectrum. It is also strange that iron(II), iron(III) and copper(II) addition results in the changes in the UV-Vis spectra of ligand (which is indicative of complex formation, I would say), but changes nothing in the emission spectrum. It should be explained somehow.

2. Depending on the ligand, cobalt(II) could turn into low-spin and high-spin state [doi: 10.1007/b95411] since it has an electron configuration of 3d7. Moreover, the mixtures of low- and high-spin states could exist (including complexes with Schiff bases) [doi: 10.1007/b95411]. In the presence of an oxidizer, low-spin cobalt(II) ion tends to give up the lone electron occupying the dx2–y2 orbital, thus, transforming into low-spin Co3+. It happened that these states can be easily distinguished using 1H NMR spectroscopy since both states of Co2+ are paramagnetic while low-spin of Co3+ is diamagnetic: low-spin cobalt(II) complex shows very broad bands and relatively small paramagnetic shifts (like, from -20 to 20 ppm [doi: 10.1021/ic00139a030]), high-spin complex of cobalt(II) shows narrow peaks and significant paramagnetic shift [doi: 10.1007/s00723-008-0161-1], while low-spin cobalt(III) complex being diamagnetic shows no paramagnetic shift. Therefore, glancing over your Figure 7, I can state that you have complex of cobalt(III) in low-spin state. You can compare with the results of e.g. paper [doi: 10.1080/00958972.2018.1512708, 10.1134/S0036023620010209]. Therefore, the mechanism of Co2+ detection can be in oxidation of cobalt(II), and the reduced ligand changes its fluorescence spectrum.

3. I do not believe that it is a good idea to approximate the titration curves with three straight lines. It actually cannot be: titration curve is always a fragment of logistic curve which can be fitted by straight line only in the beginning and the end. To obtain the equilibrium constants, these data should rather be processed using optimization procedure (maximum likelyhood approach, using e.g. https://k-ev.org/). Graphical methods are outdated and should better be avoided when stoichiometry or equilibrium constant is to be determined.

4. It is a very good point about dissociation of ligand in the presence of weak acid anions. The same had been reported previously [doi: 10.1080/00319104.2020.1774878]. I believe that other anions of weak acid (OH-, in particular) are also capable of making the same effect. I believe, this conclusion should be formulated in more clear manner.